# Effect of Age and Gender on Taste Function as Measured by the Waterless Empirical Taste Test

**DOI:** 10.3390/diagnostics13203172

**Published:** 2023-10-11

**Authors:** Rong-San Jiang, Yi-Fang Chiang

**Affiliations:** 1Department of Otolaryngology, Taichung Veterans General Hospital, Taichung 407219, Taiwan; rsjiang@vghtc.gov.tw; 2Department of Medical Research, Taichung Veterans General Hospital, Taichung 407219, Taiwan; 3School of Medicine, College of Medicine, National Yang Ming Chiao Tung University, Taipei 112304, Taiwan

**Keywords:** age, self-administered Waterless Empirical Taste Test, gender, tastant, taste

## Abstract

The effect of age and gender on taste function is rarely investigated. Therefore, we tried to study the effect of age and gender on taste function as evaluated by the Waterless Empirical Taste Test (WETT^®^). The WETT^®^ consists of 40 strips that are coated with one of five tastants (sucrose, citric acid, sodium chloride, caffeine, or monosodium glutamate). Each tastant is prepared with four different concentrations. These 40 strips are interspersed with an additional 13 tasteless strips. To implement the WETT^®^, a strip was placed on the middle portion of the tongue. The subjects closed their mouth and tasted the strip. They then chose one of six answers (sweet, sour, salty, bitter, brothy, or no taste at all). If the answer was correct, one score was acquired. One-hundred-and-twenty healthy men and women were collected in this study. Among them, there were 40 subjects in each age group of 20–39 years, 40–59 years, and ≥60 years. The overall taste and individual tastant function decreased with age, particularly between subjects aged 20–39 years and those aged ≥60 years. The overall taste and individual tastant function were better in females than in males, especially for subjects aged older than 59 years. Our results show that taste function is affected by both age and gender.

## 1. Introduction

Taste is an often-overlooked sense [1]. According to a 2016 nationwide survey in the US, 26.3 million persons aged 40 years and older are estimated to have gustatory problems [2]. This problem may cause anxiety, depression, or nutritional deficiencies in these individuals [1]. Therefore, taste dysfunction has become an important issue. More and more studies have been performed to study taste function.

There is great physiological variability in taste function among healthy subjects [3]. It is generally believed that smell sensitivity in the human population declines with age [4]. On the other hand, the influence of age on taste function is thought to be small and variable [5]. The effect of aging on taste function has been previously explored in a systemic review, with the conclusion being that taste sensitivity declines with age, although the extent of decline was not the same across different studies [6]. Regarding the aspect of gender’s effect on taste, it has been suggested that taste preference, detection ability, and response to tastants have gender differences [7]. However, the exact nature of these sex differences remains undetermined [7]. Gudziol and Hummel [8] studied taste function in a European population and found that the female taste function was more sensitive than that seen in males. In another study, gender was found to affect the perception of sour and bitter tastes [9]. In contrast, Yong et al. [10] found that neither age nor gender had an effect on taste function in 90 healthy Chinese adults; however, only subjects <65 years were included in their study.

In order to investigate the taste issue, correct measurement of a person’s taste function is mandatory. Solution-based taste tests have been used to measure gustatory function [11,12]. Although these tests have been considered to be reliable in evaluating gustatory function, they have several limitations in wide applications. Solution-based taste tests are usually time-consuming, need self-preparation, and require an assistant to perform the test [13]. Therefore, several non-solution-based tests are developed to overcome the drawbacks of solution-based tests [13]. These non-solution-based tests use tablets, edible wafers, or taste strips to measure taste function [14,15,16]. One such test, the Waterless Empirical Taste Test (WETT^®^, Sensonics International, Haddon Heights, NJ, USA), is currently available on the market. The WETT^®^ is comprised of a series of disposable plastic strips whose tastants include monosodium glutamate (umami taste), in addition to the traditional ones of sucrose, citric acid, sodium chloride, and caffeine. It has been validated and found to be as sensitive or more sensitive to age and sex than solution-based taste tests [17,18].

In our previous study, we investigated the effect of age and gender on taste function using traditional solution-based taste tests [19]. Our results showed that both age and gender affected taste function to some extent. The correct identification ability of tastants decreased with age, but the influences were not uniform. The ability to identify sweet and salty tastes was not affected by age. Compared with the results of other studies, the effect of age and gender on taste function is still controversial. One of the reasons for this may be the differences in the methods of taste test used across different studies. The WETT^®^ is a non-solution-based taste function and is more convenient and simpler to use. Moreover, the WETT^®^ adds an umami taste in addition to the traditional four tastes [17,18]. Because the WETT^®^ is commercially available, it is important to understand the effect of age and gender on its test results. Therefore, its results can be interpreted more appropriately. In this study, we aimed to investigate the effect of age and gender on taste function using the WETT^®^.

## 2. Materials and Methods

### 2.1. Study Subjects

Healthy Taiwanese volunteers were enrolled in this study. Inclusion criteria were good health, aged between 20 and 90 years, and normal self-rated smell and gustatory function. Exclusion criteria were previous surgery on wisdom teeth, middle ear surgery, and acute oral infections. Anyone who had a history of neurological or psychological diseases was also excluded from the study.

In order to investigate the effect of age and sex on taste function, we divided eligible subjects into male and female groups. In the male and female groups, we further classified subjects into 3 age groups. These 3 age groups were as follows: 20–39 years, 40–59 years, and ≥60 years. In total, there were 6 subject groups. Each group included 40 volunteers (Figure 1). The WETT^®^ was performed to measure their taste function. The Institutional Review Board (II) of Taichung Veterans General Hospital approved the study (Protocol code: CE21257B). All subjects who attended this study provided written informed consent.

### 2.2. Taste Test

The WETT^®^ consists of 53 disposable plastic strips. Among them, 40 were tastant strips, and another 13 were tasteless strips (Figure 2). At the end of each plastic strip, there was a monomer cellulose pad. In the tastant strips, the pad contained one of 5 tastants (sucrose, citric acid, sodium chloride, caffeine, or monosodium glutamate). Each tastant had four different concentrations. The concentrations of sucrose were 0.20, 0.10. 0.05, and 0.025 g/mL; the concentrations of citric acid were 0.20, 0.10. 0.05, and 0.025 g/mL; the concentrations of sodium chloride were 0.25, 0.125. 0.0625, and 0.0313 g/mL; the concentrations of caffeine were 0.088, 0.044. 0.022, and 0.011 g/mL; and the concentrations of monosodium glutamate were 0.135, 0.068. 0.034, and 0.017 g/mL. Therefore, there were 20 different tastant strips. In each test, these 20 tastant strips were presented twice to each subject in a counter-balanced order. In contrast, the pad on the tasteless strip was comprised only of monomer cellulose without any tastant. The 13 tasteless strips were interspersed among the tastant strips in a specific pattern in order to negate the need for rinsing the mouth as in other taste tests.

When performing the WETT^®^, the subject took the strip one by one, and placed the pad of the strip on the middle portion of the anterior tongue. Then, the mouth was closed, and the subject tasted the pad [20]. About 10 s later, the subject chose one of 6 answers (sweet, sour, salty, bitter, brothy [umami], or no taste at all) to indicate what they had tasted (Figure 3). If the subject answered correctly for tastant strips, one score was given. The scores of the 40 tastant strips were summed as the total correct score for the 5 tastants. The total score ranged from 0 to 40. The correct score foreach individual tastant was also calculated. The score for each tastant ranged from 0 to 8. It usually required10 to 15 min for each subject to complete the WETT^®^.

### 2.3. Statistical Analyses

In this study, the Mann–Whitney U test was used to compare male and female ages. The Mann–Whitney U test was performed to compare the total correct scores for the 5 tastants and the correct scores for each individual tastant in each age group between the male and female subjects. The Kruskal–Wallis test was undergone to compare the total correct scores for the 5 tastants and correct scores for each individual tastant among the 3 age groups. The normative data for the WETT^®^ were set at the 10th percentile score. SPSS version 17.0 (SPSS, Inc., Chicago, IL, USA) was used to perform all computations. We defined two-tailed *p*-values < 0.05 as statistically significant.

## 3. Results

### 3.1. Study Subjects

In this study, 240 healthy Taiwanese volunteers were enrolled. Among them, there were 40 male subjects between the age of 20 to 39 years, 40 male subjects between the age of 40 to 59 years, 40 male subjects aged ≥60 years, 40 female subjects aged 20–39 years, 40 female subjects aged 40–59 years, and 40 female subjects aged ≥60 years (Figure 1). For the male subjects, the ages ranged from 22 to 39 with a mean ± standard deviation age of 26.9 ± 4.72, from 40 to 59 with a mean ± standard deviation age of 47.5 ± 6.06, and from 60 to 80 with a mean ± standard deviation age of 68.3 ± 5.69 for the ≥60-year-oldgroup. The female age ranged from 22 to 39 with a mean ± standard deviation age of 29.6 ± 5.10, from 40 to 59 with a mean ± standard deviation age of 49.4 ± 6.05 for the, and from 60 to 83 with a mean ± standard deviation age of 68.4 ± 5.19 for the ≥60-year-old group. Age was not significantly different between the males and females for the three age groups (*p* = 0.896, 0.175, 0.866, respectively).

### 3.2. Waterless Empirical Taste Test

The total correct score for five tastants and the correct score for each individual tastant are shown in Table 1.

For the total five tastants, the score ranged from 14 to 39 with a mean ± standard deviation score of 28.7 ± 6.1 for male subjects and from 14 to 40 with a mean ± standard deviation score of 29.4 ± 6.1 for female subjects in the 20 to 39-year-old group. The score ranged from 6 to 38 with a mean ± standard deviation score of 25.3 ± 7.8 for male subjects and from 7 to 38 with a mean ± standard deviation score of 26.8 ± 7.6 for female subjects in the 40 to 59-year-old group. The score ranged from 6 to 37 with a mean ± standard deviation score of 21.5 ± 8.6 for male subjects and from 12 to 39 with a mean ± standard deviation score of 27.1 ± 6.5 for female subjects in the ≥60-year-old group.

For the sweet tastant, the score ranged from 0 to 8 with a mean ± standard deviation score of 5.3 ± 2.2 for male subjects and from 0 to 8 with a mean ± standard deviation score of 5.4 ± 2.0 for female subjects in the 20 to 39-year-old group. The score ranged from 0 to 8 with a mean ± standard deviation score of 4.4 ± 2.5 for male subjects and from 0 to 8 with a mean ± standard deviation score of 4.9 ± 2.2 for female subjects in the 40 to 59-year-old group. The score ranged from 0 to 8 with a mean ± standard deviation score of 3.4 ± 2.4 for male subjects and from 1 to 8 with a mean ± standard deviation score of 4.9 ± 1.9 for female subjects in the ≥60-year-old group.

For the sour tastant, the score ranged from 0 to 8 with a mean ± standard deviation score of 6.9 ± 1.6 for male subjects and from 0 to 8 with a mean ± standard deviation score of 7.0 ± 1.6 for female subjects in the 20 to 39-year-old group. The score ranged from 1 to 8 with a mean ± standard deviation score of 6.4 ± 1.8 for male subjects and from 1 to 8 with a mean ± standard deviation score of 6.5 ± 1.7 for female subjects in the 40 to 59-year-old group. The score ranged from 0 to 8 with a mean ± standard deviation score of 5.2 ± 2.1 for male subjects and from 3 to 8 with a mean ± standard deviation score of 6.1 ± 1.8 for female subjects in the ≥60-year-old group.

For the salty tastant, the score ranged from 1 to 8 with a mean ± standard deviation score of 5.7 ± 1.7 for male subjects and from 1 to 8 with a mean ± standard deviation score of 6.0 ± 1.9 for female subjects in the 20 to 39-year-old group. The score ranged from 0 to 8 with a mean ± standard deviation score of 5.6 ± 2.2 for male subjects and from 1 to 8 with a mean ± standard deviation score of 5.9 ± 2.1 for female subjects in the 40 to 59-year-old group. The score ranged from 0 to 8 with a mean ± standard deviation score of 5.0 ± 2.2 for male subjects and from 2 to 8 with a mean ± standard deviation score of 5.8 ± 1.6 for female subjects in the ≥60-year-old group.

For the bitter tastant, the score ranged from 2 to 8 with a mean ± standard deviation score of 6.7 ± 1.6 for male subjects and from 3 to 8 with a mean ± standard deviation score of 6.6 ± 1.6 for female subjects in the 20 to 39-year-old group. The score ranged from 0 to 8 with a mean ± standard deviation score of 4.7 ± 2.8 for male subjects and from 0 to 8 with a mean ± standard deviation score of 5.7 ± 2.3 for female subjects in the 40 to 59-year-old group. The score ranged from 0 to 8 with a mean ± standard deviation score of 4.2 ± 2.5 for male subjects and from 0 to 8 with a mean ± standard deviation score of 5.3 ± 2.3 for female subjects in the ≥60-year-old group.

For the umami tastant, the score ranged from 0 to 8 with a mean ± standard deviation score of 4.0 ± 2.8 for male subjects and from 0 to 8 with a mean ± standard deviation score of 4.5 ± 2.6 for female subjects in the 20 to 39-year-old group. The score ranged from 0 to 8 with a mean ± standard deviation score of 4.2 ± 2.9 for male subjects and from 0 to 8 with a mean ± standard deviation score of 3.9 ± 2.6 for female subjects in the 40 to 59-year-old group. The score ranged from 0 to 8 with a mean ± standard deviation score of 3.9 ± 3.1 for male subjects and from 0 to 8 with a mean ± standard deviation score of 5.1 ± 2.7 for female subjects in the ≥60-year-old group.

Figure 4 shows the comparison of the total correct scores for the five tastants and the correct score for each individual tastant in the three age groups between the males and females. Younger subjects (age 20–39 years) tended to have higher scores than older subjects (age ≥60 years) in all tastants, except brothy. Younger male subjects had significantly better taste function than older male subjects for the five tastants (*p* = 0.001), including the sweet tastant (*p* = 0.002), sour tastant (*p*< 0.001), and bitter tastant (*p*< 0.001). Younger male subjects also had significantly better taste function than male adult subjects (age 40–59 years) for the bitter tastant (*p* = 0.002). For females, younger subjects had significantly better taste function than older people for both the sour tastant (*p* = 0.039) and bitter tastant (*p* = 0.023).

Females tended to have higher scores than males in all tastants and all age groups, particularly for older people (age ≥60 years) (Table 1). Older female subjects had significantly better taste function than older male subjects for the sweet and bitter tastants.

The normative data of the WETT^®^ in a Taiwanese population were defined as the score at the 10th percentile (Table 2). For all subjects aged 20–39 years, the 10thpercentile score was 22 for all five tastants, 3 for the sweet tastant, 5 for the sour tastant, 3 for the salty tastant, 4 for the bitter tastant, and 0 for the umami tastant. For all subjects aged 40–59 years, the 10thpercentile score was 16 for all five tastants, 1 for the sweet tastant, 4 for the sour tastant, 2 for the salty tastant, 1 for the bitter tastant, and 0 for the umami tastant. For male subjects aged ≥60 years, the 10^th^ percentile score was 9 for all five tastants, 0 for the sweet tastant, 3 for the sour tastant, 2 for the salty tastant, 1 for the bitter tastant, and 0 for the umami tastant. For female subjects aged ≥60 years, the 10thpercentile score was 19 for all five tastants, 3 for the sweet tastant, 3 for the sour tastant, 3 for the salty tastant, 2 for the bitter tastant, and 1 for the umami tastant.

## 4. Discussion

The administrative procedures that were used for the WETT^®^ in this study are similar to those performed in one of the traditional solution-based taste tests: the whole-mouth suprathreshold test [19]. In contrast, the whole-mouth suprathreshold test only uses four tastant solutions, including sucrose, citric acid, sodium chloride, and caffeine. It does not include the brothy tastant. However, five concentrations of the solutions are prepared for all four tastants. Powders of sucrose, citric acid, sodium chloride, and caffeine are dissolved in distilled water to prepare five concentrations of four tastant solutions. The concentrations of the sweet solution are 0.08, 0.16, 0.32, 0.64, and 1.28 mol/L of sucrose. The concentrations of the sour solution are 0.0026, 0.0051, 0.0102, 0.0205, and 0.0410 mol/L of citric acid. The concentrations of the salty solution are 0.032, 0.064, 0.128, 0.256, and 0.512 mol/L of sodium chloride. The concentrations of the bitter solution are 0.0026, 0.0051, 0.0102, 0.0205, and 0.0410 mol/L of caffeine. In total, there are 20 different tastant solutions used in the whole-mouth suprathreshold test.

At the onset of the whole-mouth suprathreshold test, a small cup with 10 mL of 1 of 20 different tastant solutions is presented to the subject. The order of presentation of the tastant solutions is in a counter-balanced order. The solution in the cup is sipped, swished in the mouth for 10 s, and expectorated. The subject is then asked to choose one of the four answers(sweet, sour, salty, or bitter)to indicate the taste of the solution. If they cannot decide, a best guess must be made. One score is acquired if a correct identification of the taste is made. In the whole-mouth suprathreshold test, the subject is asked to rinse their mouth with distilled water between successive cups. The 20 different tastant solutions are tested twice. The score of a whole-mouth suprathreshold test ranged from 0 to 40. Usually, it took about 30 min to complete a whole-mouth suprathreshold test.

In our previous study investigating the effect of age and gender on taste function using the whole-mouth suprathreshold test [17], younger subjects (aged 20–39 years) tended to have higher scores than those who were older (aged ≥60 years) in all tastants except the sweet tastant. Younger male subjects had significantly better taste function than older males for all four tastants (*p*< 0.001), sour tastant (*p*< 0.001), and bitter tastant (*p* = 0.004). Younger male subjects also had significantly better taste function than older male adult subjects (age 40–59 years) for all four tastants (*p* = 0.01). For females, the younger subjects had significantly better taste function than those who were older for all four tastants (*p* = 0.033) and the sour tastant (*p* = 0.001). Older female subjects also displayed significantly worse taste function than younger female adult subjects (age 40–59 years) for the sour tastant (*p* = 0.02). Whether using the traditional whole-mouth suprathreshold test or the current WETT^®^, both tests demonstrated that taste function decreased with age, but the decrease varies between individual tastants.

Regarding the whole-mouth suprathreshold test, the score was not significantly different between males and females of the same age group for the sour, salty, and bitter tastants, but for the sweet tastant, the score was significantly higher for females than males in the 40–59 year age group (*p* = 0.047) [19]. In the current study involving the WETT^®^, females tended to have higher scores than males in all tastants and for all age groups, particularly for older people (age ≥ 60 years). Older female subjects displayed significantly better taste function than older male subjects for the sweet and bitter tastants. It seemed that the WETT^®^ was more sensitive to gender than the whole-mouth suprathreshold test.

The taste function has been compared between Chinese and American healthy adults using the WETT^®^ [21]. The results of the WETT^®^ showed that there were no significant differences in the total, sweet, and salty scores between the Chinese and American subjects. However, the Chinese subjects had a 28.40% higher score for the umami taste than the American subjects, but the American subjects had 24.12 and 21.79% higher scores for the bitter and sour tastes than the Chinese subjects, respectively. This indicated that race influenced taste function as well. Therefore, the normative data of the WETT^®^ need to be established for different populations. Taste function has rarely been studied in Asian people. In this study, we tried to establish the normative data of the WETT^®^ for Taiwanese people. However, more subjects need to be included in the study to establish more correct normative data in the future.

In addition to ethnicity, multiple other factors, such as genetics, might also influence taste function [22]. Individual taste preferences may be partially genetically determined [23]. Recently, it has been shown that the individual ability to taste bitter compounds, such as phenylthiocarbamide(PTC) and propylthiouracil(PROP), has a bimodal distribution. One well-known bitter taste receptor is T2R38. Human T2R38 functionality is altered by several polymorphisms in the TAS2R38 gene [24]. Among them, two polymorphisms(positions 49, 262, and 296) are common; one encodes a functional T2R38 and the other encodes a nonfunctional T2R38 [25]. The functional T2R38 contains proline (P), alanine (A), and valine (V) residues, and the nonfunctional T2R38 contains alanine (A), valine (V), and isoleucine (I) at these positions, respectively [26]. Homozygous PAV/PAV individuals are “supertasters” that perceive PTC and PROP as intensely bitter [25]. Heterozygote PAV/AVI individuals are intermediate tasters. On the other hand, homozygous AVI/AVI individuals are “nontasters”. The supertasters, intermediate tasters, and nontasters can be differentiated using PTC/PROP taste strips (Sensonics International, Haddon Heights, NJ, USA). In this study, we did not use PTC/PROP taste strips to test our subjects to see whether they are supertasters, intermediate tasters, or nontasters. Therefore, the responses to the bitter tastant in our subjects might be biased.

## 5. Conclusions

The WETT^®^ is a newly developed taste test. It possesses the advantages of being more convenient to both prepare and perform, requiring neither liquid tastants nor liquid rinses, as well as being self-administrative. In this study, it has been demonstrated that age and gender influenced taste function and that taste function decreased with age, with females seeming to possess better taste function than males. However, the effect of age and gender on taste function was different between individual tastants. Additionally, we tried to establish the normative data of the WETT^®^ for Taiwanese people. However, more subjects need to be included in the study to establish more correct normative data in the future.

## Figures and Tables

**Figure 1 diagnostics-13-03172-f001:**
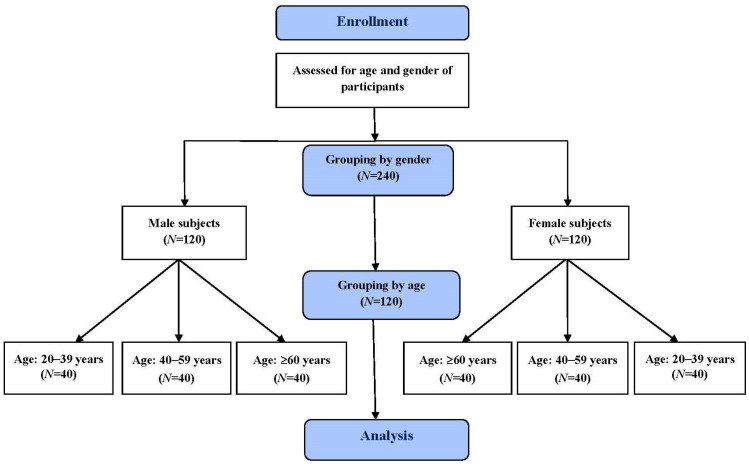
Participant flowchart.

**Figure 2 diagnostics-13-03172-f002:**
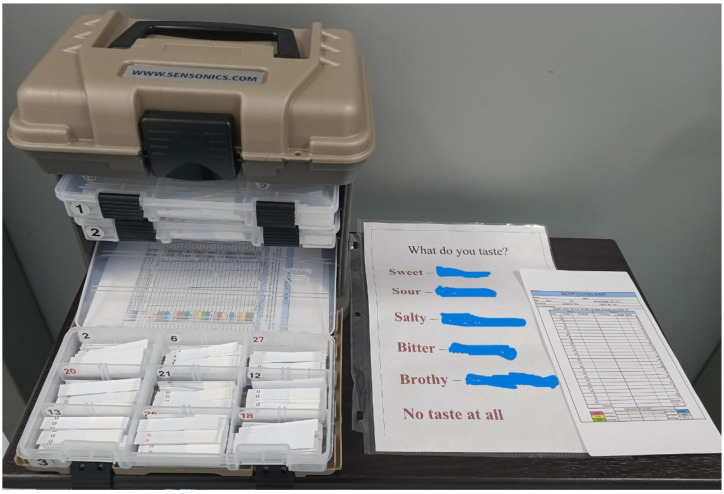
Waterless Empirical Taste Test.

**Figure 3 diagnostics-13-03172-f003:**
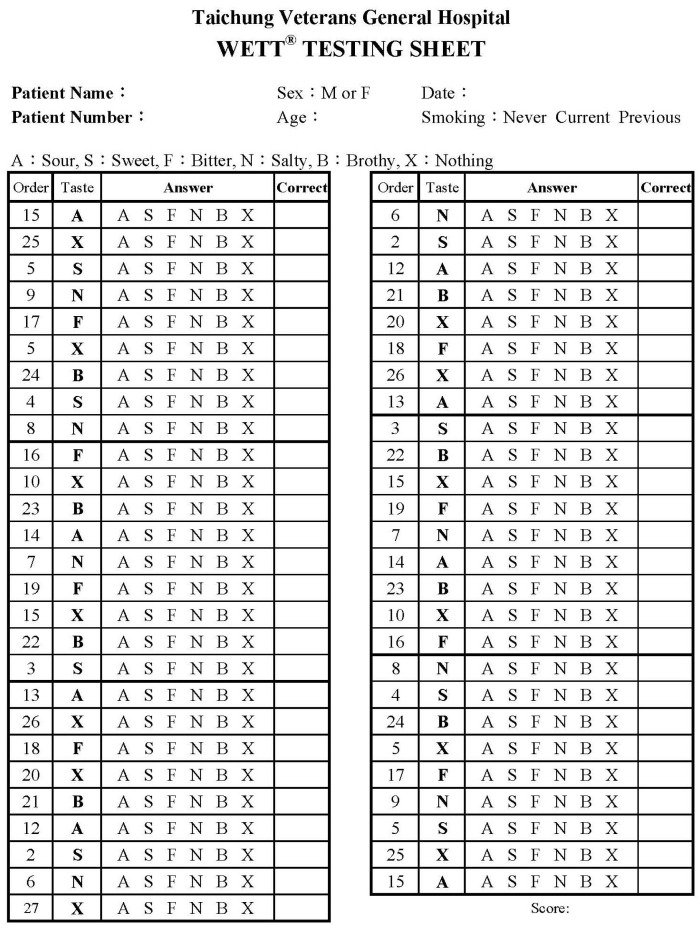
WETT^®^ testing sheet shows the presenting sequence of tastes.

**Figure 4 diagnostics-13-03172-f004:**
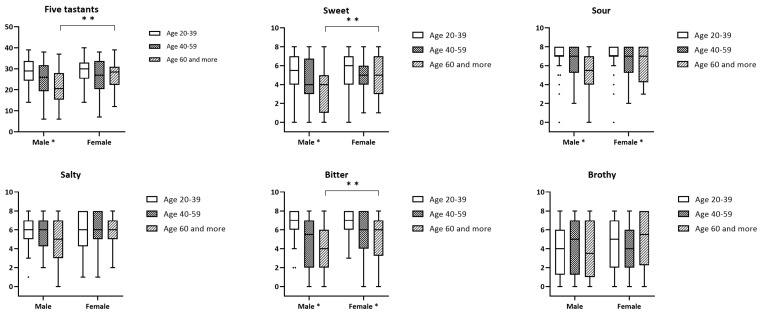
The total correct scores for the 5 tastants and correct score for each individual tastantwere shown in the plots. The 25th, 50th, and 75th quartiles were depicted in the boxes. The range of score was indicated by the lines extending parallel from the boxes. * indicates *p* < 0.05; ** indicates *p* < 0.01.

**Table 1 diagnostics-13-03172-t001:** Total correct scores for the 5 tastants and correct score for each individual tastant.

	Male		Female		*p*
	Mean	±SD	Mean	±SD	
Total score for 5 tastants					
Age 20–39	28.7	(6.1)	29.4	(6.1)	0.661
Age 40–59	25.3	(7.8)	26.8	(7.6)	0.353
Age 60+	21.5	(8.6)	27.1	(6.5)	0.002
*p*	0.001		0.263		
Sweet tastant					
Age 20–39	5.3	(2.2)	5.4	(2.0)	0.957
Age 40–59	4.4	(2.5)	4.9	(2.2)	0.390
Age 60+	3.4	(2.4)	4.9	(1.9)	0.005
*p*	0.003		0.431		
Sour tastant					
Age 20–39	6.9	(1.6)	7.0	(1.6)	0.849
Age 40–59	6.4	(1.8)	6.5	(1.7)	0.893
Age 60+	5.2	(2.1)	6.1	(1.8)	0.069
*p*	<0.001		0.042		
Salty tastant					
Age 20–39	5.7	(1.7)	6.0	(1.9)	0.389
Age 40–59	5.6	(2.2)	5.9	(2.1)	0.580
Age 60+	5.0	(2.2)	5.8	(1.6)	0.085
*p*	0.216		0.738		
Bitter tastant					
Age 20–39	6.7	(1.6)	6.6	(1.6)	0.781
Age 40–59	4.7	(2.8)	5.7	(2.3)	0.133
Age 60+	4.2	(2.5)	5.3	(2.3)	0.037
*p*	<0.001		0.027		
Umami tastant					
Age 20–39	4.0	(2.8)	4.5	(2.6)	0.465
Age 40–59	4.2	(2.9)	3.9	(2.6)	0.638
Age 60+	3.9	(3.1)	5.1	(2.7)	0.061
*p*	0.956		0.104		

**Table 2 diagnostics-13-03172-t002:** The normative data of WETT^®^ in a Taiwanese population.

	5 Tastants	Sweet	Sour	Salty	Bitter	Umami
Male subjects						
Age 20–39	21.1	2	5	3	4.1	0
Age 40–59	14.2	1	4	2	0	0
Age 60+	9	0	2.1	2	1	0
Female subjects						
Age 20–39	21.1	3	5.1	3	4	1
Age 40–59	16.1	2	4.1	2.1	2.1	0
Age 60+	18.1	2.1	3	3	2	1
All subjects						
Age 20–39	21.1	3	5	3	4	0
Age 40–59	16	1	4	2	0.1	0
Age 60+	13.1	1	3	3	1	0
All male subjects	14	0.1	3	2.1	1	0
All female subjects	18	2	4	3	2.1	1

## Data Availability

Not applicable.

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
