# Peer review of "Effect of Age and Gender on Taste Function as Measured by the Waterless Empirical Taste Test"

_diagnostics, 2023, doi:10.3390/diagnostics13203172_

Round 1

Reviewer 1 Report

The manuscript was clearly written and easy to understand.

The rationale of conducting this study should be further strengthened or explained. As the authors mentioned, the effect of age and sex on taste function was already reported by their prior studies and other researchers' studies. So what is the main point here? 

The authors mentioned at the end of the Discussion that "In this study, we tried to establish the normative data of WETT® for Taiwanese. However, more subjects need to be included in the study to establish a more correct normative data in the future". But in the Conclusion they said "we also established the normative data of WETT® for a Taiwanese population". These two sections are contradictory.

For taste science papers, please use "umami" to name the 5th taste instead of "brothy". The company developing WETT also used the term "umami".

Did the authors screen and exclude subjects who were supertasters? It could be done with propylthiouracil (PROP). If this was not done, it could be mentioned as a limitation in the Discussion section.

Author Response

Dear Editor,

Thanks very much for handling our manuscript.

We are thankful the opportunity to revise our manuscript to address the comments raised during the editorial review.

In this version, we have modified the manuscript accordingly to reviewers’ comments. Moreover, we added extra citations of different authors in order to lower the self-citation rate. The changes have been highlighted in red in the revised manuscript.

-------------------------------------------------------------------------------------------------------
Reviewer 1

The manuscript was clearly written and easy to understand.

The rationale of conducting this study should be further strengthened or explained. As the authors mentioned, the effect of age and sex on taste function was already reported by their prior studies and other researchers' studies. So what is the main point here? 

Response: Thanks for the suggestion. In our previous study, we have investigated the effect of age and sex on taste function using the traditional solution-based taste tests. As compared with the results of other studies, the effect of age and sex on taste function is still controversial. One of the reasons may be due to differences in the methods of taste test used across different studies. The Waterless Empirical Taste Test is a non-solution-based taste function and is more convenient and simple to use. Moreover, the Waterless Empirical Taste Test adds unami taste in addition to the traditional 4 tastes. Because the Waterless Empirical Taste Test is commercially available, it is important to understand the effect of age and sex on its test results. Therefore, its results can be interpreted more appropriately.

The authors mentioned at the end of the Discussion that "In this study, we tried to establish the normative data of WETT® for Taiwanese. However, more subjects need to be included in the study to establish a more correct normative data in the future". But in the Conclusion they said "we also established the normative data of WETT® for a Taiwanese population". These two sections are contradictory.

Response: Thanks for the suggestion. We changed the statement “we also established the normative data of WETT® for a Taiwanese population” in the Conclusion to the same statement “We tried to establish the normative data of WETT® for Taiwanese. However, more subjects need to be included in the study to establish a more correct normative data in the future.” as in the section of Discussion to avoid the contradictory statements.

For taste science papers, please use "umami" to name the 5th taste instead of "brothy". The company developing WETT also used the term "umami".

Response: Thanks for the suggestion. We changed all "brothy" description to "umami".

Did the authors screen and exclude subjects who were supertasters? It could be done with propylthiouracil (PROP). If this was not done, it could be mentioned as a limitation in the Discussion section.

Response: Thanks for the suggestion. We did not screen and exclude subjects who were supertasters in this study. We added a paragraph to mention this as a limitation.

Reviewer 2 Report

Though the paper has new interesting traits few points need to be well-considered and developed:

1 Inclusion and exclusion criteria are missing

2 Neurological evaluation, taste as well smell are strictly connected and are a CNS affair

3 Therefore, what about those individuals who can't perform successfully? Should we consider any cerebral lesion? Or are they smokers or drinkers?

4 The meter of judge and therefore of setting towards a final conclusion should be supported by more solid shreds of evidence...

5 is there a real control group?

Best regards

Though the paper has new interesting traits few points need to be well-considered and developed:

1 Inclusion and exclusion criteria are missing

2 Neurological evaluation, taste as well smell are strictly connected and are a CNS affair

3 Therefore, what about those individuals who can't perform successfully? Should we consider any cerebral lesion? Or are they smokers or drinkers?

4 The meter of judge and therefore of setting towards a final conclusion should be supported by more solid shreds of evidence...

5 is there a real control group?

Best regards

Author Response

Dear Editor,

Thanks very much for handling our manuscript.

We are thankful the opportunity to revise our manuscript to address the comments raised during the editorial review.

In this version, we have modified the manuscript accordingly to reviewers’ comments. Moreover, we added extra citations of different authors in order to lower the self-citation rate. The changes have been highlighted in red in the revised manuscript.

-------------------------------------------------------------------------------------------------------

Reviewer 2

Though the paper has new interesting traits few points need to be well-considered and developed:

1 Inclusion and exclusion criteria are missing.

Response: We appreciate the comments from reviewer. We have added the inclusion and exclusion criteria as follows: Inclusion criteria were good health, age between 5 and 90 years, and normal self-rated gustatory function. Exclusion criteria were previous surgery on wisdom teeth, middle ear surgery, and acute oral infections. Anyone who had a history of neurological or psychological diseases was also excluded from the study.

2 Neurological evaluation, taste as well smell are strictly connected and are a CNS affair

Response: We appreciate the comments from reviewer. Although we did not perform a neurological or psychological test on our healthy volunteers, we did ask about their history of neurological and psychological diseases. Anyone who had a history of neurological or psychological diseases was excluded from the study.

3 Therefore, what about those individuals who can't perform successfully? Should we consider any cerebral lesion? Or are they smokers or drinkers?

Response: We appreciate the comments from reviewer. In this study, no subject can’t perform the Waterless Empirical Taste Test successfully. We did not exclude any outliner from the statistical analysis. No cerebral lesion has been suspected in all subjects by asking their medical histories.

4 The meter of judge and therefore of setting towards a final conclusion should be supported by more solid shreds of evidence.

Response: We appreciate the comments from reviewer. We have added a paragraph to discuss the limitations of our study.

5 is there a real control group?

Response: We appreciate the comments from reviewer. We only included healthy volunteers in this study.

Round 2

Reviewer 1 Report

The authors have addressed my concerns adequately.

Author Response

Dear Editor,

Thanks very much for handling our manuscript.

We are thankful the opportunity to revise our manuscript to address the comments raised during the editorial review.

In this version, we have modified the manuscript accordingly to reviewers’ comments.

-------------------------------------------------------------------------------------------------------
Reviewer 1

The authors have addressed my concerns adequately.

Response: Thanks for your kind suggestions on this manuscript.

Reviewer 2 Report

I recognize the effort of the authors.

However, without a control group, the overall results of this study are a little deficient.

Recently, .It has line 279

Author Response

Dear Editor,

Thanks very much for handling our manuscript.

We are thankful the opportunity to revise our manuscript to address the comments raised during the editorial review.

In this version, we have modified the manuscript accordingly to reviewers’ comments.

-------------------------------------------------------------------------------------------------------

Reviewer 2

I recognize the effort of the authors.

However, without a control group, the overall results of this study are a little deficient.

Response: We appreciate the comments from reviewer. In the future, we shall collect patients with taste dysfunction and test them with the Waterless Empirical Taste Test to be a control group.

Recently, .It has line 279

Response: Thanks for the reviewer’s correction. We have corrected the errors.